# Not All Languages Are Created Equal in LLMs: Improving Multilingual Capability by Cross-Lingual-Thought Prompting

Haoyang Huang[1]*, Tianyi Tang[2]*, Dongdong Zhang[1]†, Wayne Xin Zhao[2]
Ting Song[1], Yan Xia[1], Furu Wei[1]
[1]Microsoft Research Asia, China
[2]Gaoling School of Artificial Intelligence, Renmin University of China
https://github.com/microsoft/unilm

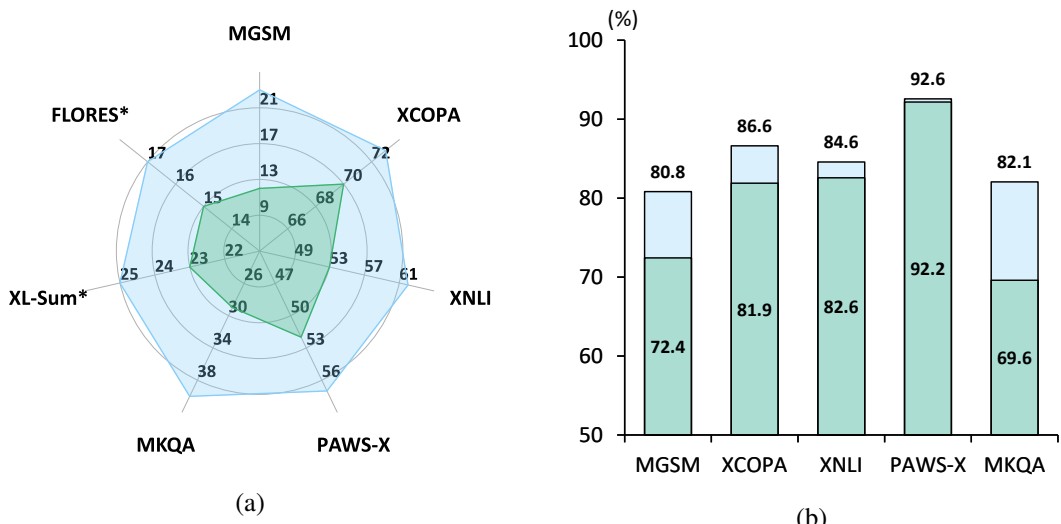

Figure 1: Comparing the effectiveness of the Cross-Lingual-Thought prompt versus the baseline basic prompt on 7 representative benchmarks covering 27 languages: (a) Enhancing the multilingual capability of `text-davinci-003` under the zero-shot learning, and (b) Narrowing the gap between the average performance and the best performance of each task in different languages.

## Abstract

Large language models (LLMs) demonstrate impressive multilingual capability, but their performance varies substantially across different languages. In this work, we introduce a simple yet effective method, called *cross-lingual-thought prompting* (**XLT**), to systematically improve the multilingual capability of LLMs. Specifically, XLT is a generic template prompt that stimulates cross-lingual and logical reasoning skills to enhance task performance across languages. We conduct comprehensive evaluations on 7 typical benchmarks related to reasoning, understanding, and generation tasks, covering both high-resource and low-resource languages. Experimental results show that XLT not only remarkably enhances the performance of various multilingual tasks but also significantly reduces the gap between the average performance and the best performance of each task in different languages. Notably, XLT brings over 10 points of average improvement in arithmetic reasoning and open-domain question-answering tasks.

## 1 Introduction

Large language models (LLMs) demonstrate impressive multilingual capability in a wide range of natural language processing tasks, including language generation, knowledge utilization, and complex reasoning (Zhao et al., 2023). Their performance in downstream tasks has been shown to reach or even surpass human-level performance (Brown et al., 2020; Chowdhery et al., 2022; Scao et al., 2022). The capabilities of LLMs stem from the extensive volume of training data they leveraged (Kaplan et al., 2020). The training data for current models is primarily dominated by the English language corpus, but it also encompasses data from other languages, as described in GPT-3 (Brown et al., 2020), PaLM (Chowdhery et al., 2022), and BLOOM (Scao et al., 2022), *etc*.

There are over 7,000 languages worldwide, with the vast majority being low-resource or extremely low-resource languages (Forkel et al., 2022). Despite the latest GPT-4 model (OpenAI, 2023) demonstrating some generalization capabilities in

*Equal contribution. † Corresponding author.

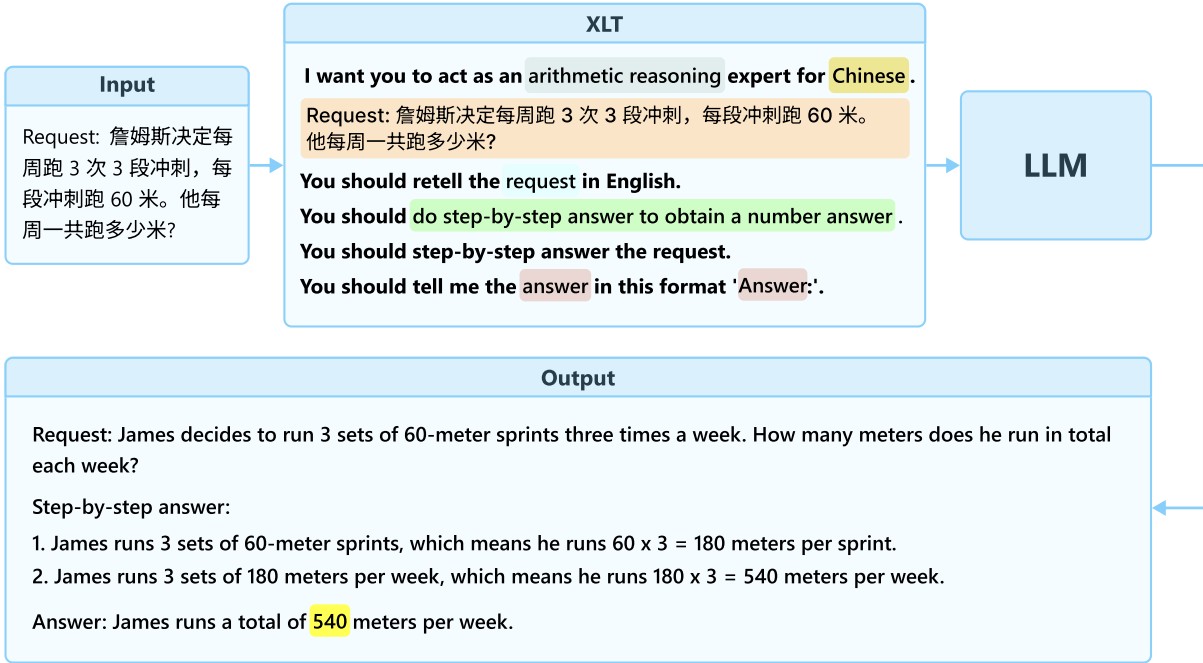

Figure 2: Overview of our method. Given a request, its associated meta information is filled into the placeholders of the XLT template to form the language-independent prompt, which is fed to the LLM to enhance the generation of responses in the desired format.

multilingual tasks as evaluated on the MMLU benchmark (Hendrycks et al., 2021), it is still the case that LLMs do not have equal capability to handle all languages, leading to imbalanced capability across different languages. Furthermore, several evaluation results (Bang et al., 2023; Jiao et al., 2023; Hendy et al., 2023; Zhu et al., 2023) indicate that large models struggle with understanding and generating non-English languages, particularly in low-resource or extremely low-resource languages. Therefore, to democratize language intelligence and minimize performance gaps in different language, it is essential and meaningful to stimulate and enhance the multilingual capability of models in non-English and low-resource languages.

Intuitively, LLMs can improve multilingual capability by augmenting data (Lin et al., 2022) or fine-tuning models (Chen et al., 2021, 2022), but both are computationally expensive. Alternatively, in-context learning with prompts can also boost performance (Brown et al., 2020; Ahuja et al., 2023; Wei et al., 2022c) but is limited to monolingual tasks (Sanh et al., 2022).

This work explores a universal in-context learning approach to enhance the multilingual capability of LLMs. We introduce a simple yet effective method, called cross-lingual-thought prompting (**XLT**), to enable models to handle various natu-

ral language processing tasks across different target languages. Our method employs a generic and language-independent prompt, which eliminates the need to update model parameters. Depending on the task input type, cross-lingual-thought prompting guides the large language model to assume the role of an expert in a specific language for a particular task. Given its predefined meta information, XLT directs LLMs to respond logically through a process involving problem understanding, cross-lingual thinking, task analysis, task execution, and output formatting. During this process, our method is designed to stimulate models' cross-lingual and logical reasoning skills, enabling them to respond to input requests regardless of the language. For enhanced performance, few-shot learning can also be employed with our method by providing an LLM-generated response output as a demonstration using cross-lingual-thought prompting zero-shot learning.

We conduct a comprehensive evaluation to verify the effectiveness of XLT across seven representative multilingual benchmarks of natural language reasoning, understanding, and generation tasks. Each benchmark includes multilingual data covering both high-resource and low-resource languages. The experimental results demonstrate that our method can significantly improve the perfor-

```
I want you to act as a task_name expert for task_language .
 task_input
You should retell/repeat the input_tag in English.
You should task_goal .
You should step-by-step answer the request.
You should tell me the output_type ( output_constraint ) in this format ' output_type :'.
```

Figure 3: Illustration of XLT template. Referring to Figure 2 and Appendix for instantiated examples.

mance of all benchmarks across languages under both zero-shot and few-shot learning settings. Notably, XLT achieves an average gain of over 10 points on the MGSM and MKQA benchmarks. Furthermore, we observe that our prompting method significantly reduces the gap between the average performance and the best performance of each task in different languages, indicating its potential to democratize language intelligence.

## 2 Cross-Lingual-Thought Prompting

Although LLMs are capable of accepting any input and generating responses, users typically structure their requests in the form of prompts to elicit the desired output. The design of these prompts is crucial for achieving optimal performance on downstream tasks, as LLMs are sensitive to the format of the prompts chosen (Zhao et al., 2021). Through a process called instruction tuning (Wei et al., 2022a), models can develop the ability to follow natural language instructions (Wei et al., 2022b), which can reduce their sensitivity to prompt engineering (Wei et al., 2022a). In accordance with the guidelines of the OpenAI cookbook[1], we propose a cross-lingual thought prompting template, denoted as the XLT template. This generic template allows LLMs to respond to requests with cross-lingual thought and supports a wide range of multilingual tasks.

Figure 3 displays the XLT template, with the colored sections representing placeholders. Figure 2 showcases an example of instantiated prompt for the Chinese request. The following section will explain the details of constructing XLT.

### 2.1 Construction of XLT

The XLT template is designed to emulate the process humans employ when handling multilingual tasks. Our template is written in English, as English is the dominant language during LLM pre-

training, and existing research indicates that English prompting is more effective for multilingual tasks (Shi et al., 2023). In contrast to the vanilla prompt that only includes a task description, our XLT template aims to elicit multilingual capability through cross-lingual thoughts. This template comprises six logical instructions in sequence. To complete the template, only seven placeholders need to be filled in based on intrinsic knowledge of the task and the request, as depicted in igure 3.

**Role Assigning** . First, the model receives a role definition that helps establish the model's behavior. This concept is akin to the system role of Chat-GPT[2]. To achieve this, we simply need to fulfill the task name with a known category (such as commonsense reasoning or paraphrase identification), along with the language of the task in the task language field.

**Task Inputting** . Second, we explicitly append the request as the task input . The request is basically structured in terms of the task type so as to make sure the model can comprehend it. For example, in the natural language inference task, the two sentence inputs are specified with "premise" and "hypothesis", respectively.

**Cross-lingual Thinking** . We encourage the model to engage in cross-lingual thought by rephrasing the requested content in English, which is the dominant language used as a pivot language by Shi et al. (2023) and Ahuja et al. (2023). Rephrasing the requested content enclosed in the input tag helps the model better understand the request in its native language and knowledge. Our observations suggest that using keywords such as "retell" or "repeat" while rephrasing the content may result in better performance in practice.

[1]https://github.com/openai/openai-cookbook

[2]https://platform.openai.com/docs/guides/chat/introduction

**Task Analyzing** . After rephrasing the task input, we need to complete the task in task goal . This step is comparable to the task description used in conventional prompting methods. In practice, we can get the task information from the literature or seek assistance from ChatGPT to generate effective prompts for solving the task (Jiao et al., 2023).

**CoT Task Solving** . We then ask the model to follow the instructions and complete the task step by step. Since LLMs exhibit a strong ability to maintain a chain-of-thought (Wei et al., 2022c), we carefully design instructions to guide the model, with the hope that it will respond to our instructions in a step-by-step manner and utilize the intermediate outputs to aid in solving the task.

**Output Formatting** . Finally, we should regularize the output format of the model to obtain the exact answer. LLMs are utilized in a zero- or few-shot manner, and they tend to generate texts that may not conform to the format of the target answer. Fortunately, LLMs possess a strong ability to follow instructions, and we can define the output format in terms of output type and output constraint . The output type can be a number, index, or text, while the output constraint is optional and determined based on the task requirements. Output constraint may include length limitations, language specifications, and other relevant factors.

## 2.2 XLT for Few-shot Learning

The above construction of XLT can be directly fed to LLMs to yield outputs, which is performed in the zero-shot learning setting. In addition, we also explore incorporating demonstrations into XLT to enable few-shot learning. Different from previous work that just appends model outputs to the corresponding request (Shi et al., 2023) or utilizes a verbalizer to format the output, our method constructs the demonstrations with better formatted model outputs from a step-by-step processing-based XLT. As illustrated in Figure 4, we first sample a few examples from the development set and incorporate the requested parts into XLT. The zero-shot learning is performed over LLM to collect responses that are further aligned with those of the samples. Only response-aligned requests are assembled with the corresponding model responses to form final demonstrations for few-shot learning. In this way, the demonstrations are constructed with rich logical knowledge via XLT, which will cater to the XLT-based generation of new requests. In practice,

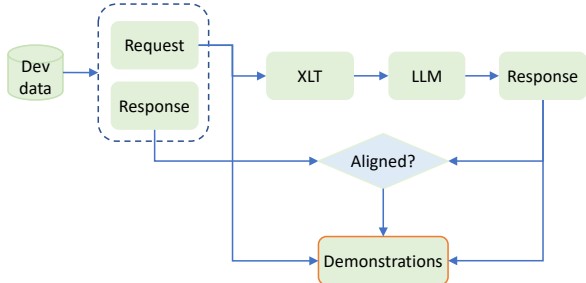

Figure 4: Construction process for few-shot learning.

we can also correct or design the demonstrations for better alignment with the instruction logic.

## 3 Experiments

To comprehensively verify the effectiveness of our method on language-independent generality, we evaluate our XLT template on different LLMs covering various natural language processing tasks in multiple languages.

### 3.1 Experimental Setups

#### 3.1.1 Tasks and Benchmarks

We conduct evaluations on seven typical benchmarks related to reasoning, understanding, and generation tasks that can represent different capabilities of LLMs, encompassing both high-resource and low-resource languages. These benchmarks cover 27 different languages, including English (en), German (de), Russian (ru), French (fr), Chinese Simplified (zh), Spanish (es), Japanese (ja), Italian (it), Vietnamese (vi), Turkish (tr), Indonesian (id), Swahili (sw), Arabic (ar), Korean (ko), Greek (el), Thai (th), Bulgarian (bg), Hindi (hi), Estonian (et), Bengali (bn), Tamil (ta), Galician (gl), Urdu (ur), Telugu (te), Javanese (jv), Haitian Creole (ht), and Southern Quechua (qu). In terms of the language distribution statistics in the Common Crawl Monthly Archives[3] and the language performance of LLMs (Shi et al., 2023; Ahuja et al., 2023), we have arranged them in the order of language frequency from high-resource to low-resource. In particular, the frequency of some underrepresented languages is even less than 0.1% (*e.g.,* bn, ta, gl, ur, te, jv, ht, qu).

- **Reasoning tasks**

– **Arithmetic Reasoning**. The MGSM (Shi et al., 2023) benchmark contains grade school mathe-

---

[3] https://commoncrawl.github.io/cc-crawl-statistics/plots/languages

matical problems and asks the model to calculate the correct answer. It covers 11 languages, and we utilize the accuracy score for evaluation.

– **Commonsense Reasoning**. The XCOPA (Ponti et al., 2020) benchmark contains one premise and two choices. It asks the model to choose which one is the result or cause of the premise. It covers 11 languages from 11 diverse families, and we utilize the accuracy score for evaluation.

• **Understanding tasks**

– **Natural Language Inference**. The XNLI (Conneau et al., 2018) benchmark contains one premise and one hypothesis and requires the model to determine whether the hypothesis is entailed, contradicted, or neutral conditioned on the premise. It covers 15 languages, and we utilize the accuracy score for evaluation.

– **Paraphrase Identification**. The PAWS-X (Yang et al., 2019) benchmark contains two sentences and requires the model to judge whether they paraphrase each other or not. It covers 7 languages, and we utilize the accuracy score for evaluation.

• **Generation tasks**

– **Question Answering**. The MKQA (Longpre et al., 2021) benchmark contains an open-domain question and asks the model to predict a short answer. Since it has unanswerable questions or long questions that do not have precise answers, we remove these questions during evaluation. It covers 25 languages, and we choose a subset of 10 languages, including de, en, es, fr, ja, ru, th, tr, vi, and zh. We utilize the token overlap F1 score for evaluation.

– **Summarization**. The XL-Sum* (Hasan et al., 2021) (250 test samples randomly sampled from XL-Sum per language) benchmark contains a long news article and wants the model to summarize it into a short text. It covers 44 languages, and we choose a subset of 6 languages, including en, es, fr, tr, vi, and zh. We utilize the ROUGE-1 score (Lin, 2004) for evaluation.

– **Machine Translation**. The FLORES* (Costa-jussà et al., 2022) (200 test samples randomly sampled from FLORES-200 per language) benchmark contains parallel text from Wikimedia projects for 204 languages, yielding over 40,000 translation directions. We choose a subset of 12 directions, including high resource to high resource translation (*i.e.,* zh↔ ru and de↔vi), high

resource to low resource translation (*i.e.,* zh↔ th and zh↔jv), and low resource to low resource translation (*i.e.,* th↔gl and jv↔th). We utilize the SacreBLEU score (Papineni et al., 2002; Post, 2018) for evaluation.

Among these benchmarks, MGSM, XCOPA, XNLI, PAWS-X, and MKQA are parallel, *i.e.,* the instances are semantics-equivalent across each language. For all benchmarks, we report the results on the test sets using all instances (Table 5), except for XL-Sum and FLORES-200, where we only sample 250 and 200 examples respectively to show the trend of generation performance. In the few-shot setting, we randomly choose examples from the development set if they have, otherwise, we translate the English training set into corresponding languages to construct several examples.

### 3.1.2 Baselines

**Basic Prompt** are the vanilla in our experiments that were proposed and suggested in previous work. After determining the prompt, we format each monolingual instance using the English basic prompt. This setting is similar to the monolingual prompting in MEGA (Ahuja et al., 2023). The basic prompts used for the evaluation of each benchmark are listed in Table 5. Note that, we dismiss the baseline using native-language, since MEGA (Ahuja et al., 2023) reveals monolingual prompting is superior to cross-lingual prompting.

**Chain-of-Thought** (CoT) prompting invokes LLMs to generate a series of intermediate results to solve reasoning tasks (Wei et al., 2022c), which is still effective under multilingual scenarios (Shi et al., 2023). In experiments, we append the instruction *"Let's think step-by-step and tell me the answer in the end"* after the input to prompt LLMs.

**Translate-English** leverages the robust capabilities of LLMs in English to tackle multilingual tasks, as suggested by both Shi et al. (2023) and Ahuja et al. (2023). This approach translates instances from other languages into English beforehand. In practice, we utilize the Google Translate API to translate examples into English and apply the basic prompt to format them. Note that, we do not apply this method to generation tasks since they require the output in respective language rather English.

**XLT** utilizes the proposed template consisting of multiple instructions introduced in Section 2. The

instantiated XLT templates for each benchmark are listed in Table 6.

In few-shot learning scenarios, for basic prompt, we use the same template as an additional input to the model. For XLT, we provide the exemplars with XLT template inputs and anticipate desirable step-by-step outputs as outlined in Figure 4. In the subsequent evaluation, we apply the 5-shot setting, except for the XL-Sum* experiments, which use the 3-shot setting due to input length constraints.

### 3.1.3 LLMs

We mainly evaluate two LLMs from the GPT-3.5 series models:

- `text-davinci-003`[4] is trained using instruction tuning and reinforcement learning from human feedback (Ouyang et al., 2022). It can perform a wide range of natural language tasks with satisfactory results.

- `gpt-3.5-turbo`[4] is optimized for chat based on `text-davinci-003` and suitable for traditional NLP tasks. It is the most capable GPT-3.5 model.

To verify the compatibility of our XLT template, we further incorporate LLaMA-2-Chat (Touvron et al., 2023) (`Llama-2-70b-chat-hf`) as our base models. It is an open-source model that has been trained through supervised fine-tuning and reinforcement learning from human feedback on the base LLaMA 2 model. In addition, we also refer to the existing results from other LLMs, such as `code-davinci-002`[4], when the evaluation is comparable. During inference, we employ greedy search (*i.e.,* temperature=0) to generate the LLM responses. We find LLMs have excellent instruction-following abilities to respond to our instructions in the given format. Therefore, we just extract the part after "Answer format:" as labels.

### 3.2 Experimental Results

**Multilingual Capability.** We comprehensively evaluate XLT's performance over seven tasks. The average score of `text-davinci-003` is summarized in Figure 1(a) and Table 1, and more details are listed in Appendix A. As for the CoT prompting, it can enhance reasoning tasks while becomes less effective on understanding and generation tasks. In terms of the Translate-En prompting, it can boost the performance in the zero-shot settings while

[4]https://platform.openai.com/docs/models/gpt-3-5

may not work well in the few-shot settings. Overall, compared to the three baseline methods, XLT achieves significant improvements over two LLMs for all tasks on both zero-shot and few-shot settings regardless of the language difference, except for a slight drop on the PAWS-X benchmark in the zero-shot setting. It is noted that XLT achieves remarkable gains of nearly 20 points on average in the MGSM benchmark for the arithmetic reasoning task and around 10 points on average in the MKQA benchmark for the open-domain question answering task. The experiments demonstrates the effectiveness of XLT for empowering LLM with multilingual capability.

As for the compatibility test, we list the results of LLaMA-2-Chat on the MGSM benchmark in Table 7. It is notable that LLaMA 2 can also benefit from our cross-lingual-thought, which further demonstrates the generality of our XLT template. However, the gains of LLaMA-2-Chat is not as good as GPT-based models. Our analysis reveals this gap can primarily be attributed to LLaMA 2's poorer multi-step instruction-following ability.

**Language Democratization.** Furthermore, we try to assess the democratization degree of tasks between languages by defining a "democratization score", which calculates the average percentage of performance attained by different languages relative to the best performance among all languages. Given the evaluation scores of $s_1, s_2, \ldots, s_l$ corresponding to $l$ language on a task, the democratization score is formulated as:

$$\frac{\sum_{i=1}^{l} s_i}{l} / \max\{s_i\}_{i=1}^{l}. \tag{1}$$

Table 2 presents the degree of democratization for tasks across languages under both zero-shot learning and few-shot learning, and we further summarize it in Figure 1(b) by averaging all scores per task regardless of the setting and model differences. We can observe that XLT leads to higher democratization scores in general, particularly for XCOPA, and MKQA. As for MGSM, XNLI, and PAWS-X, our XLT can improve performance in multiple languages, where the overall performance of the baseline is consistently lower but the gap between languages is smaller as shown in Tables 7, 9, and 10. In conclusion, our method can reduce the performance gap between languages and improve the language democratization of LLMs.

| Settings | | Reasoning | | Understanding | | Generation | | |
|---|---|---|---|---|---|---|---|---|
| | | MGSM | XCOPA | XNLI | PAWS-X | MKQA | XL-Sum* | FLORES* |
| Zero-shot | `text-davinci-003` | | | | | | | |
| | **Basic Prompt** | 12.5 | 70.1 | 53.3 | 52.0 | 29.0 | 23.7 | 15.4 |
| | **CoT** | 25.7 | 70.9 | 53.0 | **57.8** | 30.9 | 23.8 | 15.8 |
| | **Translate-En** | 15.7 | 68.0 | 54.8 | 55.0 | – | – | – |
| | **XLT** | **23.9** | **73.3** | **62.4** | 57.1 | **40.2** | **25.2** | **17.7** |
| | `gpt-3.5-turbo` | | | | | | | |
| | **Basic Prompt** | 23.3 | 76.9 | 52.6 | 65.5 | 31.6 | 24.7 | 19.1 |
| | **CoT** | 45.5 | 78.3 | 54.8 | 61.0 | 14.8 | 25.4 | 19.7 |
| | **Translate-En** | 27.1 | 75.7 | 52.2 | **66.8** | – | – | – |
| | **XLT** | **70.0** | **80.3** | **65.5** | 63.6 | **42.7** | **26.1** | **21.2** |
| Few-shot | `text-davinci-003` | | | | | | | |
| | **Basic Prompt** | 45.5 | 75.6 | 59.1 | 68.7 | 39.1 | 26.8 | – |
| | **Translate-En** | 46.5 | 77.4 | 56.9 | 68.5 | – | – | – |
| | **XLT** | **55.4** | **81.3** | **67.5** | **72.2** | **49.6** | **27.3** | – |
| | `gpt-3.5-turbo` | | | | | | | |
| | **Basic Prompt** | 63.0 | 80.1 | 61.4 | 66.4 | 43.7 | 25.5 | – |
| | **Translate-En** | 65.1 | 81.9 | 58.3 | 63.7 | – | – | – |
| | **XLT** | **72.5** | **85.9** | **65.0** | **69.1** | **52.5** | **27.9** | – |

Table 1: The average scores in different languages for the seven benchmarks in zero-shot and few-shot settings. We omit the results (denoted as "–") of Translate-En since it is not applicable for generation tasks.

| Settings | | Reasoning | | Understanding | | Generation |
|---|---|---|---|---|---|---|
| | | MGSM | XCOPA | XNLI | PAWS-X | MKQA |
| | | *Zero-shot setting* | | | | |
| `text-davinci-003` | | | | | | |
| **Basic Prompt** | | 65.2 | 77.8 | 83.8 | **97.1** | 60.2 |
| **CoT** | | 65.4 | 80.1 | 83.5 | 89.5 | 61.4 |
| **Translate-En** | | **77.2** | 78.7 | **86.0** | 95.3 | 51.6 |
| **XLT** | | 68.5 | **82.1** | 80.7 | 88.4 | **78.7** |
| `gpt-3.5-turbo` | | | | | | |
| **Basic Prompt** | | 73.0 | 83.6 | 80.5 | 89.0 | 61.8 |
| **CoT** | | 66.7 | 85.7 | 80.7 | 88.9 | 46.4 |
| **Translate-En** | | 80.4 | 84.6 | 79.8 | 90.7 | 54.1 |
| **XLT** | | **84.1** | **89.1** | **88.0** | **96.2** | 75.3 |
| | | *Few-shot setting* | | | | |
| `text-davinci-003` | | | | | | |
| **Basic Prompt** | | 75.4 | 82.0 | 82.5 | 88.2 | 74.3 |
| **Translate-En** | | 77.1 | 82.6 | 79.5 | 87.8 | 68.5 |
| **XLT** | | **84.5** | **85.6** | **85.3** | **91.6** | **82.7** |
| `gpt-3.5-turbo` | | | | | | |
| **Basic Prompt** | | 76.1 | 84.1 | 83.6 | 94.4 | 82.1 |
| **Translate-En** | | 78.6 | 86.4 | 79.2 | **95.4** | 71.3 |
| **XLT** | | **86.2** | **89.7** | **84.3** | 94.1 | **83.1** |

Table 2: The democratization degree of tasks against languages.

## 3.3 Further Analysis

In this section, we further investigate the factors that affect the performance of XLT and how they affect various multilingual benchmarks.

### 3.3.1 Ablation of XLT

For the XLT variants, we mainly conduct experiments to compare the following strategies:

- **Ablating the instructions.** Since our XLT consists of six logical instructions, we disable the *Role Assigning*, *Cross-lingual Thinking*, and *CoT Task Solving* instructions separately to analyze the contribution per instruction.

- **Reordering the instructions.** Considering the logicality of our instructions, we further change the order of the instructions in XLT to explore whether LLMs will handle tasks differently and lead to different results.

- **Changing the content word.** As prompts are usually sensitive to the word choice, we verify the robustness of XLT when alternating the rephrasing keyword with "retell", "repeat", and "translate" in the cross-lingual thinking instruction.

The outcomes are presented in Table 3, indicating that XLT surpasses almost all the variants, thereby validating the effectiveness and reasonableness of our proposed XLT method.

**The effectiveness of each instruction.** The results from the "Instruction Ablation" row indicate that: (1) *Cross-lingual Thinking* yields more significant gains compared to other instructions. This suggests that the LLM's ability of cross-lingual thinking is activated, allowing it to utilize its knowledge in English to solve tasks effectively; (2) Removing *Role Assigning* from XLT impedes the model's

| Settings | | MGSM | | XNLI | | FLORES* | |
|---|---|---|---|---|---|---|---|
| | | **de** | **zh** | **hi** | **vi** | **jv→zh** | **zh→jv** |
| | **XLT** | 79.8 | 72.6 | 61.3 | 64.8 | 19.0 | 10.5 |
| **Instruction Ablation** | *w/o Role Assigning* | 76.6 | 69.2 | 57.8 | 63.9 | 16.2 | 8.8 |
| | *w/o Cross-lingual Thinking* | 75.6 | 62.0 | 56.1 | 62.2 | 13.2 | 8.2 |
| | *w/o CoT Task Solving* | 77.0 | 68.0 | 62.9 | 65.2 | 16.8 | 9.2 |
| **Instruction Order** | Swap *Role Assigning* and *Task Inputting* | 77.2 | 71.8 | 54.2 | 61.5 | 19.6 | 11.2 |
| | Swap *Role Assigning* and *Task Analyzing* | 76.8 | 70.8 | 61.0 | 64.0 | 15.8 | 8.8 |
| | Swap *Cross-lingual Thinking* and *Task Analyzing* | 79.0 | 71.2 | 59.5 | 63.4 | 16.5 | 9.7 |
| **Rephrasing Word** | *w/* retell | 79.8 | 72.6 | 61.3 | 64.8 | 18.2 | 10.3 |
| | *w/* repeat | 77.6 | 68.0 | 60.7 | 64.6 | 19.0 | 10.5 |
| | *w/* translate | 76.4 | 70.0 | 60.1 | 64.5 | 17.5 | 10.2 |

Table 3: Performance comparison across different variants of XLT. All the experiments are conducted using `gpt-3.5-turbo` under the zero-shot setting.

| Demonstration format | en | de | ru | fr | zh | es | ja | sw | th | bn | te | Avg. |
|---|---|---|---|---|---|---|---|---|---|---|---|---|
| Basic input + Basic output | 84.0 | 79.2 | 78.8 | 78.8 | 70.8 | 81.2 | 68.8 | 70.8 | 68.8 | 65.2 | 44.8 | 71.9 |
| Basic input + XLT output | 82.4 | 72.4 | 71.2 | 75.2 | 64.4 | 78.8 | 63.2 | 66.8 | 53.6 | 54.8 | 32.4 | 65.0 |
| XLT input + XLT output | 84.8 | 81.4 | 80.2 | 79.2 | 71.8 | 81.6 | 72.8 | 71.2 | 69.8 | 64.4 | 40.8 | **72.5** |

Table 4: Performance comparison across different few-shot variants on the MGSM benchmark. All the experiments are conducted with 5 demonstrations using `gpt-3.5-turbo`.

understanding of the ultimate goal for diverse multilingual tasks, highlighting the task transferability of XLT; and (3) the better performance of XLT can also be attributed to *CoT Task Solving*, which requires the model to respond to complex instructions in a step-by-step manner.

**The order of logical instructions.** The performance drop is evident when the order of our designed logical instructions is switched. When designing XLT, we have taken into account the process by which humans solve multilingual problems, and this experiment further confirms the optimum order of our XLT template. Placing the *Role Assigning* instruction later may confuse the model initially. Additionally, conducting *Cross-lingual Thinking* before *Task Analyzing* is crucial since we rely on the English task-solving abilities of LLMs to handle multilingual tasks.

**The robustness of word choice for rephrasing keywords.** We can find that different words indeed affect the performance of XLT, but it is less sensitive to the other variants. Through experimentation, we have determined that "repeat" yields better results for text summarization and machine translation, while "retell" is more suitable for the remaining five tasks. Our aim is to provide XLT with a more unified template, while still allowing users to fine-tune specific keywords for optimal

performance in their tasks.

### 3.3.2 Effectiveness of XLT Few-shot Learning

As mentioned in Section 2.2, the construction of demonstrations for XLT few-shot learning differs from the previous method. We have compared XLT and basic prompt. Here, we focus on the construction of the demonstration input-output pairs and compare various demonstrations that may be used to perform XLT few-shot learning. The illustrations can be found in Figure 5.

- **Basic prompt input + Basic prompt output:** This is the normal demonstration format used in most of the previous work.

- **Basic prompt input + XLT output:** This ablation is to separate the effect of input and output formats in the demonstration.

- **XLT input + XLT output:** This is the method that we used in this work.

Observing the experimental results presented in Table 4, we can conclude that: (1) Our XLT few-shot learning outperforms all other variants, thus confirming its effectiveness. (2) The use of normal demonstrations for XLT few-shot learning leads to a decrease in performance. (3) Merely incorporating XLT as a demonstration input without its output does not result in any improvements. (4) Consistency in the demonstration for few-shot learning

is crucial, implying that the demonstration input-output format should align better with its zero-shot learning input-output format.

## 4 Related Work

### 4.1 LLM Capability Understanding

Despite the impressive capabilities of LLMs, it is crucial to determine their impact on natural language processing tasks. Liang et al. (2022) conduct a comprehensive evaluation of LLMs from various perspectives, such as accuracy, calibration, robustness, fairness, bias, toxicity, and efficiency. Bang et al. (2023) extensively evaluate the Chat-GPT model on multiple natural language processing tasks and find that the model performs well in high-resource languages but exhibits certain limitations in low-resource and non-Latin script languages. Additionally, studies by Jiao et al. (2023) and Hendy et al. (2023) compare different GPT models with supervised models for machine translation tasks and find that GPT models have competitive translation abilities in high-resource languages but perform less effectively in low-resource languages. It is worth noting that achieving multilingual generative AI capability necessitates cross-lingual knowledge to further improve the model's performance. In this context, Ahuja et al. (2023) evaluate the multilingual task understanding ability of GPT models and attempt to enhance their task processing abilities in other languages using English knowledge. Our work also focuses on evaluating the multilingual capabilities of LLMs, including reasoning, understanding, and generative capabilities. Our evaluations indicate that LLMs exhibit differences in high-resource and low-resource abilities, which necessitates additional efforts to enhance their multilingual capability.

### 4.2 Multilingual Task Processing

Multilingual knowledge has been shown to be exploitable and transferable between languages to improve model performance (Devlin et al., 2019; Conneau et al., 2020; Raffel et al., 2020; Ouyang et al., 2021; Chi et al., 2021). While much research has been devoted to multilingual understanding tasks, multilingual generation tasks are more challenging, particularly when the target language is low-resource or non-English (Ma et al., 2021; Liu et al., 2020). Two methods can enable models to support multilingual task processing: one is training a supervised model that covers multiple languages for multilingual processing (Costa-jussà et al., 2022), and the other is training a pre-trained model and using fine-tuning to transfer knowledge among languages to achieve multilingual capability (Chen et al., 2021, 2022). However, the emergence of LLMs has made it possible to directly process multilingual tasks via in-context learning (Brown et al., 2020; Ahuja et al., 2023). These LLMs, with hundreds of billions or even trillions of parameters, require a significant amount of computation resources for training, making traditional fine-tuning methods less feasible. To improve the generative ability of LLMs, researchers explore in-context learning methods that do not require updating model parameters, such as few-shot prompting (Vilar et al., 2022), automatic prompt learning (Shin et al., 2020), task-instruction prompting (Ye et al., 2023), chain-of-thought prompting (Wei et al., 2022c), *etc*. Our work builds upon these methods and proposes an optimized, generic, and language-independent prompt to enhance the multilingual capability of LLMs.

## 5 Conclusion

This work investigates the language processing capabilities of large language models in multilingual settings and expects to develop a universal framework for handling diverse multilingual tasks. To accomplish this goal, we propose a generic prompt, referred to as XLT, to enhance the multilingual capability and reduce the performance gaps among languages in tasks related to language understanding, reasoning, and generation in non-English and low-resource languages. Although our method is generally applicable across tasks and languages, we discovered that prompting design factors such as instruction logic and word choice have explicit impacts on its effectiveness. Cross-language thinking in XLT is particularly effective. Finally, we hope this work can inspire further research to prioritize the development of generic prompting. By doing so, large language models can encompass a wider range of modalities and languages.

## Acknowledgements

Tianyi Tang and Xin Zhao are supported by National Natural Science Foundation of China under Grant No. 62222215, Beijing Natural Science Foundation under Grant No. 4222027 and L233008.

## Limitations

Due to limitations imposed by the evaluation benchmarks and OpenAI API cost, we conducted tests on 27 languages, which merely scratch the surface of the vast array of languages in the world. Besides, our XLT template is based on English. It deserves to explore whether the template written in task language can lead to better performance and how to better construct the instruction in each language. Furthermore, we only verify the effectiveness of our method on two GPT-based models (*i.e.,* `text-davinci-003` and `gpt-3.5-turbo`) and LLaMA-2-Chat. It is worthwhile to investigate the generality of our template on more models, such as BLOOM and PaLM.

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

# A Additional Experiments

## A.1 Results on Reasoning Tasks

Table 7 presents the results of the MGSM benchmark. XLT significantly improves the arithmetic reasoning capabilities of both models, particularly for `gpt-3.5-turbo` in the zero-shot setting. We hypothesize that `gpt-3.5-turbo` may have undergone supervised fine-tuning (Ouyang et al., 2022) with arithmetic reasoning samples in the chain-of-thought format, which enables XLT to activate its arithmetic reasoning ability directly. For both low-resource languages (*e.g.,* sw, th, bn, and te) and high-resource languages, XLT can further enhance the performance. Even under the few-shot setting, XLT can still significantly improve the reasoning performance of both models and reduce the performance gap for all languages. Notably, for some high-resource languages, such as de, ru, fr, and es, the performance is comparable to English.

The XCOPA benchmark results are presented in Table 8. Our XLT approach significantly enhances the performance of both models in both settings, as compared to basic prompting. In the zero-shot setting, XLT demonstrates significant improvements for relatively low-resource languages (*e.g.,* sw, th, et, ta, and ht), but it underperforms the baseline for some high-resource languages such as zh and it. In the few-shot setting, XLT brings enhancements for both high- and low-resource languages. Our findings suggest that XLT is more effective for low-resource languages, particularly for `gpt-3.5-turbo` on sw, th, ta, and ht, where it yields improvements of over 10 accuracy points.

## A.2 Results on Understanding Tasks

Table 9 presents the results of the XNLI benchmark. In the zero-shot setting, our XLT significantly outperforms the basic prompt in all languages. Additionally, when using few-shot setups on high- and low-resource languages, both `text-davinci-003` and `gpt-3.5-turbo` show significant improvements compared to the basic prompt. Specifically, for low-resource languages such as th, bg, hi, and ur, XLT achieves an average improvement of 9.4 accuracy scores for `text-davinci-003` and 5.3 accuracy scores for `gpt-3.5-turbo`. This demonstrates that XLT is effective for both models, but `text-davinci-003` has better natural language inference capabilities.

Table 10 displays the comparisons on the PAWS-X task, where XLT outperforms basic prompt in all languages, particularly for low-resource languages under the few-shot setting. We observe a slight performance drop on average in zero-shot learning compared to `gpt-3.5-turbo` for some high-resource languages (*e.g.,* en, de, and fr). Based on our analysis of intermediate outputs, we infer that the drop in performance may be due to cross-lingual thinking that alters the original meaning of the two sentences, leading to difficulties in judgment. Additionally, a comparable pattern is evident in a previous study (Ahuja et al., 2023), where non-Latin script languages (ja, zh, and ko) exhibit significantly poorer performance than English or German in the few-shot setting. Nevertheless, by demonstrating the construction of XLT, we can guide the model on how to think across different languages and effectively address the aforementioned issues.

## A.3 Results on Generation Tasks

The MKQA benchmark outcomes are listed in Table 11. Across all languages in the zero-shot and few-shot settings, the XLT template shows a significant improvement over the basic prompt. It is worth noting that `text-davinci-003` performs worse than `gpt-3.5-turbo` in this task, and we speculate that the latter is optimized for open question answering, which is common in daily chat. Additionally, our findings indicate that XLT can notably enhance the performance of under-resourced languages. XLT brings over 10 points of improvement for these languages. (*e.g.,* zh, ja, vi, and tr) This aligns with previous benchmarking studies and is particularly noteworthy in this evaluation. We suspect that high-resource and low-resource languages share the same cross-lingual thinking as English to greatly leverage the LLM's ability to solve English open-domain QA.

The results of the XL-Sum* benchmark are presented in Table 12. It can be observed that XLT outperforms the basic prompt in both zero- and few-shot settings across all languages. Additionally, the LLM model exhibits a significant improvement in generating summaries under the few-shot setting compared to the zero-shot setting. This suggests that providing fewer examples can effectively guide the model in summarizing multilingual texts. Furthermore, the few-shot results revealed an interesting finding that `text-davinci-003` performed better when `gpt-3.5-turbo` and `text-davinci-003` use basic prompt. However, once XLT is enabled,

Table 5: The basic prompt of each benchmark. #Test denotes the number of instances in the test set.

| Benchmark | #Test | Basic Prompt |
|---|---|---|
| MGSM | 250 | Request: {problem} |
| XCOPA | 500 | Here is a premise: {premise}. What is the {question}? Help me pick the more plausible option: -choice1: {choice1}, -choice2: {choice2} |
| XNLI | 5,010 | {premise} Based on previous passage, is it true that {hypothesis}? Yes, No, or Maybe? |
| PAWS-X | 2,000 | Sentence 1: {sentence1} Sentence 2: {sentence2} Question: Does Sentence 1 paraphrase Sentence 2? Yes or No? |
| MKQA | 6,758 | Answer the question in one or a few words in {target_language}: {question}? |
| XL-Sum* | 250 | Summarize this article: {article} |
| FLORES* | 200 | {source} Translate from {source_language} to {target_language}: |

Table 6: Task meta data consisting of task name, input tag, task goal, output type, and output constraint per benchmark. Detailed examples of the input for each benchmark are listed in the following part.

| Benchmark | Task name | Input tag | Task goal | Output type | Output constraint |
|---|---|---|---|---|---|
| MGSM | arithmetic reasoning | request | do step-by-step answer to obtain a number answer | answer | – |
| XCOPA | commonsense reasoning | premise and the options | do step-by-step answer to pick a choice | choice number | – |
| XNLI | natural language inference | hypothesis and the premise | judge whether the hypothesis is true, false, or undetermined given the premise. The relationship can be chosen from entailment, contradiction, and neutral | relationship | – |
| PAWS-X | paraphrase identification | sentence 1 and sentence 2 | provide a yes or no answer to the question: Does Sentence 1 paraphrase Sentence 2? | answer | choosing either yes or no |
| MKQA | question answering | question | answer the question in English in one or a few words | answer | in one or a few words in {target_language} |
| XL-Sum | multilingual summarization | entire text | think step-by-step to summarize the entire text in a maximum of two sentences | summary | into one sentence in {target_language} |
| FLORES | machine translation | source sentence | provide the {target_language} translation for the English source sentence | target translation | – |

gpt-3.5-turbo outperforms text-davinci-003, highlighting the effectiveness of our approach.

Machine translation is a special generation task where the source and target are two different languages. The experiment in this part is to verify how XLT boosts machine translation tasks. Since English has been specified as the pivot language in the cross-lingual thinking in XLT, we exclude English-centric tasks to avoid language redundancy and focus on 12 non-English translation directions in the FLORES* benchmark, which includes both high-resource and low-resource languages. As shown in Table 13, XLT achieves impressive zero-shot results for all languages compared with basic prompt. For example, it significantly improves translation quality in Chinese-to-X or X-to-Chinese. The result emphasizes that XLT will potentially transfer the knowledge of a high-resource pivot language like English to the target language. While the benefit of XLT may not be as obvious for high-to-high translations, it becomes more significant for high-to-low, low-to-high, and low-to-low translations. For instance, XLT improves the translation perfor-

mance of gpt-3.5-turbo by nearly 4.0, 2.8, and 3.3 BLEU points for th→gl, jv→zh, and zh→th translations, respectively, demonstrating its effectiveness regardless of whether the source language is high-resource or low-resource. Noticing that Hendy et al. (2023) have shown that few-shot configurations do not yield significant improvements over the zero-shot setup for translation tasks, we do not evaluate the few-shot paradigm on FLORES* in this work and leave it for future exploration.

| Settings (high→low) | en | de | ru | fr | zh | es | ja | sw | th | bn | te | Avg. |
|---|---|---|---|---|---|---|---|---|---|---|---|---|
| **Zero-shot** | | | | | | | | | | | | |
| `text-davinci-003` | | | | | | | | | | | | |
| **Basic Prompt** | 19.2 | 12.8 | 15.6 | 16.4 | 15.2 | 13.6 | 12.8 | 7.2 | 8.8 | 11.6 | 4.4 | 12.5 |
| **XLT** | 30.0 | 32.4 | 23.6 | 34.8 | 29.2 | 26.8 | 26.0 | 13.6 | 18.4 | 14.8 | 12.8 | **23.9** |
| `gpt-3.5-turbo` | | | | | | | | | | | | |
| **Basic Prompt** | 32.0 | 24.8 | 28.0 | 31.6 | 22.0 | 29.2 | 22.4 | 24.4 | 16.8 | 18.0 | 7.6 | 23.3 |
| **XLT** | 84.4 | 79.8 | 77.6 | 75.2 | 72.6 | 76.8 | 71.0 | 70.8 | 63.8 | 56.8 | 42.0 | **70.0** |
| `Llama-2-70b-chat-hf` | | | | | | | | | | | | |
| **Basic Prompt** | 58.8 | 48.0 | 47.2 | 45.6 | 39.6 | 50.4 | 39.2 | 10.0 | 13.6 | 17.2 | 5.2 | 34.1 |
| **XLT** | 60.0 | 52.8 | 52.8 | 48.8 | 42.4 | 52.0 | 39.2 | 16.4 | 18.0 | 17.6 | 10.4 | **37.3** |
| **Few-shot** | | | | | | | | | | | | |
| `code-davinci-002` (Shi et al., 2023)[*] | 53.6 | 46.4 | 48.8 | 46.4 | 47.2 | 51.6 | 44.8 | 37.6 | 41.2 | 41.2 | 42.8 | 45.6 |
| `text-davinci-003` | | | | | | | | | | | | |
| **Basic Prompt** | 60.4 | 45.6 | 51.6 | 45.6 | 38.8 | 51.6 | 37.6 | 48.8 | 30.4 | 43.6 | 46.8 | 45.5 |
| **XLT** | 65.6 | 58.0 | 57.6 | 56.8 | 53.2 | 58.0 | 54.4 | 58.8 | 42.4 | 53.2 | 51.8 | **55.4** |
| `gpt-3.5-turbo` | | | | | | | | | | | | |
| **Basic Prompt** | 82.8 | 69.2 | 71.6 | 72.4 | 46.8 | 71.2 | 56.0 | 60.0 | 44.0 | 62.4 | 56.6 | 63.0 |
| **XLT** | 84.8 | 81.4 | 80.2 | 79.2 | 71.8 | 81.6 | 72.8 | 71.2 | 69.8 | 64.4 | 40.8 | **72.5** |

Table 7: Accuracy scores on the MGSM benchmark. Shi et al. (2023)[*] utilize 6-shot learning.

| Settings (high→low) | zh | it | vi | tr | id | sw | th | et | ta | ht | qu | Avg. |
|---|---|---|---|---|---|---|---|---|---|---|---|---|
| **Zero-shot** | | | | | | | | | | | | |
| `text-davinci-003` | | | | | | | | | | | | |
| **Basic Prompt** | 85.4 | 90.0 | 69.2 | 80.6 | 83.8 | 56.4 | 66.6 | 73.0 | 53.4 | 61.6 | 50.4 | 70.1 |
| **XLT** | 85.8 | 89.2 | 76.0 | 81.0 | 86.4 | 59.2 | 67.2 | 83.4 | 55.2 | 72.2 | 50.2 | **73.3** |
| `gpt-3.5-turbo` | | | | | | | | | | | | |
| **Basic Prompt** | 90.4 | 92.0 | 83.6 | 86.6 | 88.2 | 77.0 | 70.2 | 84.0 | 57.2 | 65.2 | 51.2 | 76.9 |
| **XLT** | 87.8 | 89.8 | 87.5 | 90.2 | 89.5 | 82.0 | 78.0 | 88.4 | 64.0 | 74.6 | 51.8 | **80.3** |
| **Few-shot** | | | | | | | | | | | | |
| `code-davinci-002` (Shi et al., 2023)[*] | 93.4 | 96.6 | 86.6 | 91.2 | 91.4 | 67.4 | 84.2 | 88.8 | 55.8 | 79.6 | 52.2 | 80.7 |
| `text-davinci-003` (Ahuja et al., 2023)[†] | – | 94.6 | – | 89.8 | 93.0 | 82.8 | 84.8 | 89.6 | 87.0 | 82.8 | – | – |
| **Basic Prompt** | 90.8 | 92.2 | 80.2 | 85.2 | 90.8 | 63.6 | 69.2 | 81.8 | 53.6 | 73.2 | 51.0 | 75.6 |
| **XLT** | 94.0 | 95.0 | 87.0 | 94.0 | 92.8 | 68.4 | 79.4 | 90.4 | 59.4 | 80.8 | 53.0 | **81.3** |
| `gpt-3.5-turbo` | | | | | | | | | | | | |
| **Basic Prompt** | 91.0 | 95.2 | 86.2 | 89.0 | 88.6 | 79.2 | 73.6 | 92.0 | 58.6 | 74.2 | 53.0 | 80.1 |
| **XLT** | 92.8 | 95.8 | 90.6 | 92.2 | 90.2 | 92.6 | 85.2 | 93.0 | 70.8 | 86.0 | 56.2 | **85.9** |

Table 8: Accuracy scores on the XCOPA benchmark. (Shi et al., 2023)[*] utilize 6-shot learning. Ahuja et al. (2023)[†] utilize 8-shot learning.

| Settings (high→low) | en | de | ru | fr | zh | es | vi | tr | sw | ar | el | th | bg | hi | ur | Avg. |
|---|---|---|---|---|---|---|---|---|---|---|---|---|---|---|---|---|
| **Zero-shot** | | | | | | | | | | | | | | | | |
| `text-davinci-003` | | | | | | | | | | | | | | | | |
| **Basic Prompt** | 63.6 | 59.4 | 55.9 | 60.9 | 51.6 | 59.7 | 49.5 | 53.9 | 40.8 | 51.9 | 53.2 | 49.7 | 54.4 | 49.8 | 45.3 | 53.3 |
| **XLT** | 77.4 | 67.7 | 64.2 | 68.3 | 64.8 | 69.4 | 62.0 | 61.5 | 54.3 | 58.7 | 61.1 | 56.3 | 62.6 | 55.1 | 53.0 | **62.4** |
| `gpt-3.5-turbo` | | | | | | | | | | | | | | | | |
| **Basic Prompt** | 65.4 | 55.5 | 50.6 | 53.2 | 48.8 | 59.8 | 52.1 | 54.4 | 49.6 | 50.9 | 54.9 | 44.8 | 55.7 | 49.2 | 44.8 | 52.6 |
| **XLT** | 74.4 | 68.5 | 66.0 | 69.8 | 64.9 | 69.4 | 64.8 | 65.0 | 60.1 | 62.8 | 68.3 | 62.1 | 67.7 | 61.3 | 57.3 | **65.5** |
| **Few-shot** | | | | | | | | | | | | | | | | |
| `text-davinci-003` (Ahuja et al., 2023)[†] | 79.5 | 71.7 | 67.3 | 71.8 | 65.8 | 72.2 | 66.9 | 67.6 | 57.3 | 65.1 | 69.3 | 62.0 | 70.8 | 63.3 | 55.1 | 67.1 |
| **Basic Prompt** | 71.6 | 65.8 | 62.5 | 63.4 | 56.7 | 64.6 | 59.4 | 56.9 | 48.2 | 57.3 | 62.0 | 55.0 | 62.6 | 52.4 | 48.0 | 59.1 |
| **XLT** | 79.1 | 70.8 | 70.0 | 69.5 | 69.2 | 71.0 | 67.3 | 66.9 | 59.5 | 65.7 | 67.8 | 63.7 | 70.4 | 63.5 | 58.1 | **67.5** |
| `gpt-3.5-turbo` | | | | | | | | | | | | | | | | |
| **Basic Prompt** | 73.4 | 66.3 | 60.9 | 67.9 | 60.2 | 68.1 | 60.2 | 62.6 | 55.7 | 58.8 | 64.7 | 52.7 | 64.6 | 53.8 | 50.8 | 61.4 |
| **XLT** | 77.1 | 69.3 | 64.4 | 69.6 | 62.9 | 70.6 | 63.2 | 64.4 | 60.2 | 63.4 | 66.6 | 59.8 | 66.9 | 60.0 | 56.5 | **65.0** |

Table 9: Accuracy scores on the XNLI benchmark. Ahuja et al. (2023)[†] utilize 8-shot learning.

| Settings (high→low) | en | de | fr | zh | es | ja | ko | Avg. |
|---|---|---|---|---|---|---|---|---|
| **Zero-shot** | | | | | | | | |
| `text-davinci-003` | | | | | | | | |
| **Basic Prompt** | 53.3 | 52.9 | 50.8 | 53.6 | 53.0 | 50.3 | 50.3 | 52.0 |
| **XLT** | 64.6 | 56.0 | 55.3 | 57.4 | 56.0 | 54.6 | 55.9 | **57.1** |
| `gpt-3.5-turbo` | | | | | | | | |
| **Basic Prompt** | 73.6 | 68.3 | 68.4 | 63.4 | 69.6 | 59.7 | 55.7 | **65.5** |
| **XLT** | 65.3 | 66.1 | 64.8 | 65.5 | 63.3 | 62.4 | 57.6 | 63.6 |
| **Few-shot** | | | | | | | | |
| `text-davinci-003` | | | | | | | | |
| (Ahuja et al., 2023)[†] | 72.5 | 69.8 | 71.3 | 65.2 | 70.1 | 65.4 | 65.8 | 68.6 |
| **Basic Prompt** | 77.8 | 70.6 | 72.5 | 65.0 | 71.7 | 62.5 | 60.5 | 68.7 |
| **XLT** | 76.5 | 78.8 | 77.4 | 61.1 | 78.0 | 65.0 | 68.7 | **72.2** |
| `gpt-3.5-turbo` | | | | | | | | |
| **Basic Prompt** | 65.9 | 70.3 | 66.6 | 64.1 | 68.2 | 65.6 | 64.0 | 66.4 |
| **XLT** | 73.4 | 69.8 | 68.5 | 70.9 | 67.8 | 66.4 | 66.7 | **69.1** |

Table 10: Accuracy scores on the PAWS-X benchmark. Ahuja et al. (2023)[†] utilize 8-shot learning.

| Settings (high→low) | en | de | ru | fr | zh | es | ja | vi | tr | th | Avg. |
|---|---|---|---|---|---|---|---|---|---|---|---|
| **Zero-shot** | | | | | | | | | | | |
| `text-davinci-003` | | | | | | | | | | | |
| **Basic Prompt** | 48.1 | 33.8 | 15.9 | 34.8 | 18.2 | 34.1 | 27.7 | 23.6 | 24.0 | 29.6 | 29.0 |
| **XLT** | 51.1 | 42.3 | 27.3 | 43.0 | 36.7 | 43.3 | 46.8 | 35.8 | 37.8 | 38.1 | **40.2** |
| `gpt-3.5-turbo` | | | | | | | | | | | |
| **Basic Prompt** | 51.1 | 40.6 | 28.4 | 40.1 | 16.5 | 39.3 | 25.9 | 23.3 | 26.9 | 23.7 | 31.6 |
| **XLT** | 56.7 | 46.0 | 33.9 | 47.6 | 33.0 | 47.9 | 47.5 | 36.5 | 39.1 | 38.6 | **42.7** |
| **Few-shot** | | | | | | | | | | | |
| `text-davinci-003` | | | | | | | | | | | |
| **Basic Prompt** | 52.6 | 42.3 | 21.8 | 42.9 | 33.1 | 42.8 | 45.5 | 35.5 | 37.5 | 36.6 | 39.1 |
| **XLT** | 57.6 | 49.4 | 42.7 | 50.9 | 51.0 | 50.0 | 60.0 | 46.9 | 46.9 | 40.5 | **49.6** |
| `gpt-3.5-turbo` | | | | | | | | | | | |
| **Basic Prompt** | 53.2 | 48.6 | 31.0 | 46.1 | 40.9 | 47.9 | 51.4 | 38.5 | 40.0 | 39.3 | 43.7 |
| **XLT** | 59.6 | 52.5 | 43.8 | 53.9 | 51.9 | 54.0 | 63.2 | 49.4 | 52.1 | 44.7 | **52.5** |

Table 11: F1 scores on the MKQA benchmark. The average score is the macro average F1 score.

| Settings (high→low) | en | fr | zh | es | vi | tr | Avg. |
|---|---|---|---|---|---|---|---|
| **Zero-shot** | | | | | | | |
| `text-davinci-003` | | | | | | | |
| **Basic Prompt** | 22.2 | 26.2 | 30.8 | 25.1 | 22.0 | 15.9 | 23.7 |
| **XLT** | 24.4 | 28.2 | 32.2 | 26.0 | 22.3 | 17.9 | **25.2** |
| `gpt-3.5-turbo` | | | | | | | |
| (Lai et al., 2023) | 19.7 | 20.8 | 21.1 | 17.8 | – | 14.5 | – |
| **Basic Prompt** | 25.3 | 26.2 | 30.2 | 26.3 | 21.1 | 19.2 | 24.7 |
| **XLT** | 26.8 | 28.1 | 33.3 | 26.4 | 21.3 | 20.5 | **26.1** |
| **Few-shot** | | | | | | | |
| `text-davinci-003` | | | | | | | |
| **Basic Prompt** | 29.2 | 29.6 | 33.2 | 28.3 | 22.5 | 18.1 | 26.8 |
| **XLT** | 28.2 | 30.3 | 34.4 | 29.4 | 22.7 | 18.6 | **27.3** |
| `gpt-3.5-turbo` | | | | | | | |
| **Basic Prompt** | 25.7 | 27.2 | 30.8 | 27.8 | 21.5 | 19.7 | 25.5 |
| **XLT** | 28.5 | 29.2 | 35.0 | 28.6 | 23.7 | 22.3 | **27.9** |

Table 12: ROUGE-1 scores on the XL-Sum* benchmark.

| Settings | High-High | | | | High-Low | | | | Low-Low | | | |
|---|---|---|---|---|---|---|---|---|---|---|---|---|
| | zh-ru | | de-vi | | zh-th | | zh-jv | | th-gl | | jv-th | |
| | $\rightarrow$ | $\leftarrow$ | $\rightarrow$ | $\leftarrow$ | $\rightarrow$ | $\leftarrow$ | $\rightarrow$ | $\leftarrow$ | $\rightarrow$ | $\leftarrow$ | $\rightarrow$ | $\leftarrow$ |
| Zero-shot `text-davinci-003` | | | | | | | | | | | | |
| **Basic Prompt** | 19.8 | 24.2 | 26.5 | 24.5 | 10.2 | 11.8 | 8.1 | 14.0 | 17.9 | 12.0 | 10.0 | 6.2 |
| **XLT** | **21.6** | **24.8** | **27.4** | **24.8** | **12.6** | **16.4** | **11.1** | **18.2** | **20.7** | **14.2** | **11.7** | **9.0** |
| `gpt-3.5-turbo` | | | | | | | | | | | | |
| **Basic Prompt** | 23.3 | 25.4 | 34.1 | 29.6 | 16.6 | 18.6 | 9.1 | 16.2 | 18.1 | 18.5 | 13.0 | 7.2 |
| **XLT** | **25.3** | **25.6** | **33.3** | **31.3** | **19.9** | **19.3** | **10.5** | **19.0** | **22.1** | **21.9** | **15.9** | **10.6** |

Table 13: BLEU scores on the FLORES* benchmark.

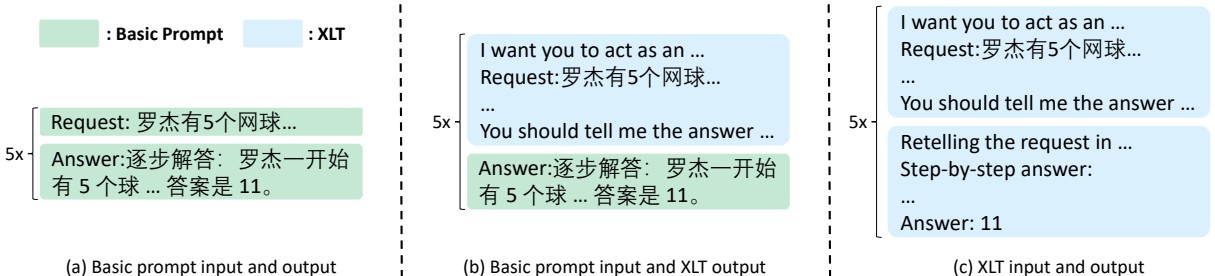

Figure 5: Illustrations of different demonstration input-output pairs in the few-shot learning.

Input:

> Request: 制作一件袍子需要 2 匹蓝色纤维布料和这个数量一半的白色纤维布料。它一共需要用掉多少匹布料?

text-davinci-003 output:

> Answer: 3 匹布料

gpt-3.5-turbo output:

> 3 匹布料。

Answer:

> 3

Figure 6: A Chinese example of the **MGSM** benchmark using **basic prompt** and the corresponding outputs under the **zero-shot** setting.

Input:

> I want you to act as an arithmetic reasoning expert for Chinese .
> Request: 制作一件袍子需要 2 匹蓝色纤维布料和这个数量一半的白色纤维布料。它一共需要用掉多少匹布料?
> You should retell the request in English.
> You should do step-by-step answer to obtain a number answer .
> You should step-by-step answer the request.
> You should tell me the answer in this format ' Answer :'.

text-davinci-003 output:

> The request is: To make a robe, you need 2 pieces of blue fiber fabric and half of that amount of white fiber fabric. How many pieces of fabric are needed in total?
> Step-by-step answer:
> 1. Start with the amount of blue fiber fabric: 2 pieces
> 2. Half of that amount is 1 piece
> 3. Add the 1 piece of white fiber fabric to the 2 pieces of blue fiber fabric
> 4. Answer: 3 pieces of fabric are needed in total.

gpt-3.5-turbo output:

> The request is asking how many pieces of fabric are needed to make a robe, given that it requires 2 pieces of blue fiber fabric and half that amount of white fiber fabric.
> To solve this problem, we need to first find out how many pieces of white fiber fabric are needed. We can do this by dividing the number of blue fiber fabric pieces by 2: 2 ÷ 2 = 1
> So we need 1 piece of white fiber fabric.
> To find the total number of fabric pieces needed, we can add the number of blue and white pieces together: 2 + 1 = 3
> Therefore, Answer: 3 pieces of fabric are needed to make the robe.

Answer:

> 3

Figure 7: A Chinese example of the **MGSM** benchmark using **XLT** and the corresponding outputs under the **zero-shot** setting.

Input:

I want you to act as an arithmetic reasoning expert for Chinese .

Request: 罗杰有5个网球。他又买了2罐网球。每罐有3个网球。他现在有多少个网球?

You should retell the request in English.

You should do step-by-step answer to obtain a number answer .

You should step-by-step answer the request.

You should tell me the answer in this format ' Answer :'.

Retelling the request in English: Roger currently has 5 tennis balls...

Step-by-step answer:

1. Roger has 5 tennis balls.

2. He buys 2 cans of tennis balls.

3. Each can has 3 tennis balls, so 2 cans will have a total of 6 tennis ...

4.To find out how many tennis balls Roger has in total, we need to add the ...

5. Adding 5 (initial tennis balls) and 6 (purchased tennis balls) gives ...

Answer: 11.

... (the remaining 4 examples)

I want you to act as an arithmetic reasoning expert for Chinese .

Request: 制作一件袍子需要 2 匹蓝色纤维布料和这个数量一半的白色纤维布料。它一共需要用掉多少匹布料?

You should retell the request in English.

You should do step-by-step answer to obtain a number answer .

You should step-by-step answer the request.

You should tell me the answer in this format ' Answer :'.

text-davinci-003 output:

Request: To make a robe, two pieces of blue fabric and half the amount of white fabric are needed. How many pieces of fabric are needed in total?
Step-by-step answer:
1. Two pieces of blue fabric are needed.
2. Half the amount of blue fabric is needed for white fabric, so that is 1 piece of white fabric (2 x 0.5 = 1).
3. To find out how many pieces of fabric are needed in total, we need to add the number of blue fabric and white fabric.
4. Adding 2 (blue fabric) and 1 (white fabric) gives us a total of 3 pieces of fabric.

Answer: 3 pieces of fabric.

gpt-3.5-turbo output:

Request: Making one robe requires 2 pieces of blue fiber fabric and half that amount of white fiber fabric. How many pieces of fabric are needed in total?
Step-by-step answer:
1. To make one robe, we need 2 pieces of blue fiber fabric.
2. We also need half that amount of white fiber fabric, which is 1 piece (2/2 = 1).
3. Therefore, to make one robe, we need a total of 2 + 1 = 3 pieces of fiber fabric.

Answer: We need 3 pieces of fiber fabric in total to make one robe.

Answer:

3

Figure 8: A Chinese example of the **MGSM** benchmark using **XLT** and the corresponding outputs under the **few-shot** setting.

Input:

```
Here is a premise: Ese oli mullikilesse mässitud. What is the cause? Help me
pick the more plausible option: -choice1: See oli õrn, -choice2: See oli väike.
```

text-davinci-003 output:

```
Choice 2 : See oli väike.
```

gpt-3.5-turbo output:

```
As an AI language model, I cannot determine the cause of the premise without
additional context. The options provided do not seem to be directly related
to the premise. Can you please provide more information or context?
```

Answer:

```
1
```

Figure 9: An Estonian example of the **XCOPA** benchmark using **basic prompt** and the corresponding outputs under the **zero-shot** setting.

Input:

```
I want you to act as a commonsense reasoning expert for Estonian .
Here is a premise: Ese oli mullikilesse mässitud. What is the cause? Help me
pick the more plausible option: -choice1: See oli õrn, -choice2: See oli väike.
You should retell the premise and the options in English.
You should do step-by-step answer to pick a choice .
You should step-by-step answer the request.
You should tell me the choice number in this format ' Choice number :'.
```

text-davinci-003 output:

```
Premise: Ese oli mullikilesse mässitud.
Options: Choice 1: See oli õrn, Choice 2: See oli väike.
Step 1: Consider the context of the premise. What kind of object was mässitud (inserted) into the pocket?
Step 2: Consider the two options. Choice 1: See oli õrn (It was soft) and Choice 1: See oli väike (It was small).
Step 3: Based on the context and the two options, the more plausible choice is Choice 2: See oli väike (It was small).
Choice number: 2
```

gpt-3.5-turbo output:

```
Premise: "Ese oli mullikilesse mässitud" (The object was wrapped in bubble wrap).
Options:
1. See oli õrn (It was fragile).
2. See oli väike (It was small).
Step 1: Based on the premise, we know that the object was wrapped in bubble wrap. It does not provide any information about the size or fragility of the item.
Step 2: However, common sense would suggest that fragile items are more likely to be wrapped in bubble wrap for protection during transportation or storage.
Therefore, the more plausible option is choice 1: "See oli õrn" (It was fragile).
Choice number: 1 .
```

Answer:

```
1
```

Figure 10: An Estonian example of the **XCOPA** benchmark using **XLT** and the corresponding outputs under the **zero-shot** setting.

Input:

I want you to act as a commonsense reasoning expert for Estonian .
Here is a premise: Mees jäi vanaks. What is the effect? Help me pick the more plausible option:
-choice1: Ta juuksed muutusid halliks, -choice2: Ta müüs oma asjad maha.
You should retell the premise and the options in English.
You should do step-by-step answer to pick a choice .
You should step-by-step answer the request.
You should tell me the choice number in this format ' Choice number :'.

Premise: A man turned on the tap.
Options: 1. The toilet filled with water. 2. Water flowed from the tap.
Step-by-step answer:
1. The premise states that a man turned on the tap, which implies that water started flowing.
2. The first option suggests that the toilet filled with water, which is possible if the tap ...
3. The second option suggests that water flowed from the tap, which is the more plausible ...
Choice Number: 2. Water flowed from the tap. The answer is choice2."

... (the remaining 4 examples)

I want you to act as a commonsense reasoning expert for Estonian .
Here is a premise: Ese oli mullikilesse mässitud. What is the cause? Help me pick the more
plausible option: -choice1: See oli õrn, -choice2: See oli väike.
You should retell the premise and the options in English.
You should do step-by-step answer to pick a choice .
You should step-by-step answer the request.
You should tell me the choice number in this format ' Choice Number :'.

text-davinci-003 output:

Premise: The object was wedged into a crack.
Options:
1. It was delicate.
2. It was small.
Step-by-step answer:
1. The premise states that the object was wedged into a crack.
2. Option 1 suggests that the object was delicate, which is not necessarily related to the object being wedged into a crack.
3. Option 2 suggests that the object was small, which is more plausible as a small object is more likely to fit into a crack.
Choice Number: 2 . It was small.The answer is choice2.

gpt-3.5-turbo output:

Premise: "The object was wrapped in bubble wrap."
Options:
1. It was fragile.
2. It was small.
Step-by-step answer:
1. The premise states that the object was wrapped in bubble wrap, which is often used to protect fragile items during transport.
2. Bubble wrap is not typically used to protect small items, so option 1 is more plausible.
3. Therefore, the more plausible option is Choice 1: It was fragile.
Choice Number: 1 . It was fragile. The answer is choice1.

Answer:

1

Figure 11: An Estonian example of the **XCOPA** benchmark using **XLT** and the corresponding outputs under the **few-shot** setting.

Input:

```
Les options ne sont pas si attrayantes. Based on previous passage, is it true
that Les choix ne sont pas trop attrayants? Yes, No, or Maybe?
```

text-davinci-003 output:

```
yes
```

gpt-3.5-turbo output:

```
maybe
```

Answer:

```
entailment
```

Figure 12: A French example of the **XNLI** benchmark using **basic prompt** and the corresponding outputs under the **zero-shot** setting. We need a verbalizer to map entailment to yes, contradiction to no, and neutral to maybe.

Input:

```
I want you to act as a natural language inference expert for French .
Premise: Les options ne sont pas si attrayantes.
Hypothesis: Les choix ne sont pas trop attrayants.
You should retell the premise and hypothesis in English.
You should judge whether the hypothesis is true (entailment), false
(contradiction), or undetermined (neutral) given the premise. The relationship
can be chosen from entailment, contradiction, and neutral .
You should step-by-step answer the request.
You should tell me the relationship in this format ' Relationship :'.
```

text-davinci-003 output:

```
Premise (in English): The options are not very attractive.
Hypothesis (in English): The choices are not too attractive.
Relationship: Contradiction
```

gpt-3.5-turbo output:

```
Premise: The options are not so attractive.
Hypothesis: The choices are not too attractive.
Relationship: Entailment . The hypothesis is true given the premise as the concepts of options and
choices are interchangeable and both refer to the same thing. The degree of attractiveness is also
conveyed in a similar manner in both statements.
```

Answer:

```
entailment
```

Figure 13: A French example of the **XNLI** benchmark using **XLT** and the corresponding outputs under the **zero-shot** setting.

Input:

```
I want you to act as a  natural language inference  expert for  French .
 Premise: savons-nous ce que nous allons dire ?
 Hypothesis: Je sais que nous n'avons aucune idée de ce que nous allons dire.
You should retell the  premise and hypothesis  in English.
You should  judge whether the hypothesis is true (entailment), false
 (contradiction), or undetermined (neutral) given the premise. The relationship
 can be chosen from entailment, contradiction, and neutral .
You should step-by-step answer the request.
You should tell me the  relationship  in this format ' Relationship :'.

 Retelled premise: do we know what we're going to say.
 Retelled hypothesis: I know we don't have a clue what we'll say.
 Relationship: contradiction

... (the remaining 4 examples)

I want you to act as a  natural language inference  expert for  French .
 Premise: Les options ne sont pas si attrayantes.
 Hypothesis: Les choix ne sont pas trop attrayants.
You should retell the  premise and hypothesis  in English.
You should  judge whether the hypothesis is true (entailment), false
 (contradiction), or undetermined (neutral) given the premise. The relationship
 can be chosen from entailment, contradiction, and neutral .
You should step-by-step answer the request.
You should tell me the  relationship  in this format ' Relationship :'.
```

text-davinci-003 output:

```
retelled premise: the options are not so attractive.
retelled hypothesis: the choices are not too attractive.
relationship:  entailment
```

gpt-3.5-turbo output:

```
retelled premise: the options are not very attractive.
retelled hypothesis: the choices are not too attractive.
relationship:  entailment
```

Answer:

```
entailment
```

Figure 14: A French example of the **XNLI** benchmark using **XLT** and the corresponding outputs under the **few-shot** setting.

Basic prompt input:

```
Sentence 1: Lito spielte für den Zingone Club Football. Sentence 2: Zingone spielte für den
Club Lito Fußball. Does Sentence 1 paraphrase Sentence 2? Yes or No?
```

text-davinci-003 output:

```
No
```

gpt-3.5-turbo output:

```
Yes
```

Answer:

```
No
```

Figure 15: A Germany example of the **PAWS-X** benchmark using **basic prompt** and the corresponding outputs under the **zero-shot** setting.

Input:

```
I want you to act as a paraphrase identification expert for Germany .
Sentence 1: Lito spielte für den Zingone Club Football
Sentence 2: Zingone spielte für den Club
Question: Does Sentence 1 paraphrase Sentence 2? Yes or No?
You should retell the  sentence 1 and sentence 2  in English.
You should provide a yes or no answer to the question: Does Sentence 1 paraphrase Sentence 2?
You should step-by-step answer the request.
You should tell me the  answer   choosing either yes or no  in this format ' Answer :'.
```

text-davinci-003 output:

```
Sentence 1: Lito played football for the Zingone Club.
Sentence 2: Zingone played football for the Lito Club.
Answer: Yes
```

gpt-3.5-turbo output:

```
Sentence 1: Lito played football for the Zingone Club.
Sentence 2: Zingone played football for the Club Lito.
Answer: No , Sentence 1 does not paraphrase Sentence 2.
Step-by-step answer:
1. Retold Sentence 1 and Sentence 2 in English.
2. Provided a yes or no answer to the question: Does Sentence 1 paraphrase Sentence 2?
3. Answered the request in the required format.
```

Answer:

```
No
```

Figure 16: A Germany example of the **PAWS-X** benchmark using **XLT** and the corresponding outputs under the **zero-shot** setting.

Input:

```
I want you to act as a  paraphrase identification  expert for  Germany .
 Sentence 1: Der Ponoru ist eine Nebenfluss des Horezu in Rumänien .
 Sentence 2: Der Ponoru ist ein Nebenfluss des Horezu in Rumänien .
 Question: Does Sentence 1 paraphrase Sentence 2? Yes or No?
You should retell the  sentence 1 and sentence 2  in English.
You should  provide a yes or no answer to the question: Does Sentence 1 paraphrase Sentence 2?
You should step-by-step answer the request.
You should tell me the  answer   choosing either yes or no  in this format ' Answer :'.

 Retelled Sentence 1: The Ponoru River is a tributary of the Horezu River in Romania .
 Retelled Sentence 2: The Ponoru River is a tributary of the Horezu in Romania .
 Answer: Yes

... (the remaining 4 examples)

I want you to act as a  paraphrase identification  expert for  Germany .
 Sentence 1: Lito spielte für den Zingone Club Football
 Sentence 2: Zingone spielte für den Club
 Question: Does Sentence 1 paraphrase Sentence 2? Yes or No?
You should retell the  sentence 1 and sentence 2  in English.
You should  provide a yes or no answer to the question: Does Sentence 1 paraphrase Sentence 2?
You should step-by-step answer the request.
You should tell me the  answer   choosing either yes or no  in this format ' Answer :'.
```

text-davinci-003 output:

```
Retelled Sentence 1: Lito played football for the Zingone Club.
Retelled Sentence 2: Zingone played football for the Lito Club.
Answer: Yes
```

gpt-3.5-turbo output:

```
Retelled Sentence 1: Lito played football for the Zingone Club.
Retelled Sentence 2: Zingone played football for the Lito Club.
Answer: No
```

Answer:

```
No
```

Figure 17: A Germany example of the **PAWS-X** benchmark using **XLT** and the corresponding outputs under the **few-shot** setting.

Input:

```
Answer the question in one or a few words in  Japanese  : 世界で一番背の高い人類は誰で
すか？
```

text-davinci-003 output:

```
最高身長者。
```

gpt-3.5-turbo output:

```
誰？
```

Answer:

```
ロバート・ワドロー
```

Figure 18: A Japanese example of the **MKQA** benchmark using **basic prompt** and the corresponding outputs under the **zero-shot** setting.

Input:

```
I want you to act as a  question answering  expert for  Japanese .
 Question: 世界で一番背の高い人類は誰ですか？
You should retell the  question  in English.
You should  answer the question in English in one or a few words.
You should step-by-step answer the request.
You should tell me the  answer   in one or a few words in Japanese  in this format ' Answer :'.
```

text-davinci-003 output:

```
Question in English: Who is the tallest human being in the world?
Answer: Robert Wadlow (2.72 m).
Answer in Japanese: 答え：ロバート・ワドロウ (2.72 m) 。
```

gpt-3.5-turbo output:

```
Retelling: Who is the tallest person in the world?
Answer: Robert Wadlow
Step-by-step response:
1. The question is asking for the tallest person in the world.
2. The answer is Robert Wadlow, who was 8 feet 11 inches (2.72 meters) tall.
Answer in Japanese: 答え：ロバート・ワドロー
```

Answer:

```
ロバート・ワドロー
```

Figure 19: A Japanese example of the **MKQA** benchmark using **XLT** and the corresponding outputs under the **zero-shot** setting.

Input:

```
I want you to act as a question answering expert for Japanese .
Question: フルハウスでスティーブが出てきたのはどのシーズンですか？
You should retell the question in English.
You should answer the question in English in one or a few words.
You should step-by-step answer the request.
You should tell me the answer in one or a few words in Japanese in this
format 'Answer :'.

Question: what season does steve come in full house?
English Answer: 5.0 seasons.
Japanese Answer: 5.0 時節

... (the remaining 4 examples)

I want you to act as a question answering expert for Japanese .
Question: 世界で一番背の高い人類は誰ですか？
You should retell the question in English.
You should answer the question in English in one or a few words.
You should step-by-step answer the request.
You should tell me the answer in one or a few words in Japanese in this
format 'Answer :'.
```

text-davinci-003 output:

```
question: who is the tallest human being in the world
english answer: sultan kösen
japanese answer: スルタン・ケーセン
```

gpt-3.5-turbo output:

```
question: who is the tallest person in the world
nenglish answer: robert wadlow
japanese answer: ロバート・ワドロー
```

Answer:

```
ロバート・ワドロー
```

Figure 20: A Japanese example of the **MKQA** benchmark using **XLT** and the corresponding outputs under the **few-shot** setting.

Input:

```
Summarize this article: Huang Ming es tan apasionado por la energía solar que construyó el llamado "Solar Valley"
(o Ciudad Solar): un conjunto de edificios que funcionan íntegramente a partir del sol. En su diseño también se
integran los paneles solares que los alimentan. Pero aunque Huang soñaba con que todo el mundo imitaría sus
esfuerzos, todavía sigue siendo la única Ciudad Solar en China (y del mundo). Video producido por Álvaro Ricciardelli.
```

text-davinci-003 output:

```
Huang Ming es un innovador apasionado por la energía solar que construyó el primer conjunto de edificios
alimentados íntegramente por el sol, conocido como "Solar Valley" o Ciudad Solar. A pesar de que Huang esperaba
que su proyecto inspirara a otros, todavía es el único de su tipo en China y el mundo.
```

gpt-3.5-turbo output:

```
Huang Ming built the "Solar Valley" in China, a collection of buildings that run entirely on solar power. Despite
his hopes that others would follow his lead, it remains the only solar city in China and the world.
```

Answer:

```
Lo llaman "El rey sol", aunque a él no le gusta.
```

Figure 21: A Spanish example of the **XL-Sum\*** benchmark using **basic prompt** and the corresponding outputs under the **zero-shot** setting.

Input:

```
I want you to act as a  multilingual summarization  expert for  Spanish .
Text: Huang Ming es tan apasionado por la energía solar que construyó el llamado "Solar Valley" (o Ciudad Solar):
un conjunto de edificios que funcionan íntegramente a partir del sol. En su diseño también se integran los
paneles solares que los alimentan. Pero aunque Huang soñaba con que todo el mundo imitaría sus esfuerzos,
todavía sigue siendo la única Ciudad Solar en China (ydel mundo). Video producido por Álvaro Ricciardelli.
You should repeat the  entire text  in English.
You should  think step-by-step to summarize the entire text in a maximum of two sentences .
You should step-by-step answer the request.
You should tell me the  summary   into one sentence in Spanish  in this format ' Summary :'.
```

text-davinci-003 output:

```
Summary:  Huang Ming es un apasionado de la energía solar que construyó la única Ciudad Solar en China y del
mundo, llamada "Solar Valley", que funciona íntegramente con paneles solares.
```

gpt-3.5-turbo output:

```
The text is in Spanish and talks about Huang Ming, who built the "Solar Valley" in China, a group of
buildings that run entirely on solar energy. Despite his hopes that others would follow his lead, it remains the
only solar city in China and the world.
To summarize in two sentences: Huang Ming built the "Solar Valley" in China, a group of buildings that run
entirely on solar energy, but it remains the only solar city in China and the world.
Summary:  Huang Ming construyó la "Ciudad Solar" en China, un conjunto de edificios que funcionan íntegramente
a partir del sol, pero sigue siendo la única ciudad solar en China y en el mundo.
```

Answer:

```
Lo llaman "El rey sol", aunque a él no le gusta.
```

Figure 22: A Spanish example of the **XL-Sum\*** benchmark using **XLT** and the corresponding outputs under the **zero-shot** setting.

Input:

```
I want you to act as a  multilingual summarization  expert for  Spanish .
 Text: Testigos del tiroteo explicaron que el atacante asaltó la mezquita a los pocos minutos de haber empezado
 la oración. La policía de Nueza Zelanda declaró que hay cuatro personas detenidas y que no puede descartarse
 que haya más implicados en los ataque. Ahora puedes recibir notificaciones de BBC Mundo. Descarga la nueva
 versión de nuestra app y actívalas para no perderte nuestro mejor contenido.
You should repeat the  entire text  in English.
You should  think step-by-step to summarize the entire text in a maximum of two sentences .
You should step-by-step answer the request.
You should tell me the  summary   into one sentence in Spanish  in this format ' Summary :'.

 The text states that a shooter attacked a mosque during prayer time in New Zealand, with four people currently
 in custody and the possibility of more involved. The article also promotes the BBC Mundo app for notifications.
 To summarize, a mosque in New Zealand was attacked during prayer time with four people currently detained and
 the potential for more involvement, while also promoting the BBC Mundo app for notifications.
 Summary: Al menos 49 personas murieron en dos tiroteos en mezquitas en Nueva Zelanda.

... (the remaining 2 examples)

I want you to act as a  multilingual summarization  expert for  Spanish .
 Text: Huang Ming es tan apasionado por la energía solar que construyó el llamado "Solar Valley" (o Ciudad Solar):
 un conjunto de edificios que funcionan íntegramente a partir del sol. En su diseño también se integran los
 paneles solares que los alimentan. Pero aunque Huang soñaba con que todo el mundo imitaría sus esfuerzos,
 todavía sigue siendo la única Ciudad Solar en China (y del mundo). Video producido por Álvaro Ricciardelli.
You should repeat the  entire text  in English.
You should  think step-by-step to summarize the entire text in a maximum of two sentences .
You should step-by-step answer the request.
You should tell me the  summary   into one sentence in Spanish  in this format ' Summary :'.
```

text-davinci-003 output:

```
Summary:  Huang Ming construyó la única Ciudad Solar del mundo, que funciona íntegramente con energía solar.
```

gpt-3.5-turbo output:

```
The text highlights Huang Ming's passion for solar energy and his creation of the "Solar Valley", a group of
buildings that run entirely on solar power, but despite his hopes for others to follow suit, it remains the
only solar city in China and the world.
Summary:  Huang Ming construyó la única Ciudad Solar en China y del mundo, un conjunto de edificios que
 funcionan íntegramente a partir del sol.
```

Answer:

```
Lo llaman "El rey sol", aunque a él no le gusta.
```

Figure 23: A Spanish example of the **XL-Sum\*** benchmark using **XLT** and the corresponding outputs under the **few-shot** setting.

Input:

Nalika ana siji vaksin pacoban katon bisa nyuda kematian Ebola, nganti saiki, ora ana obat
sing kabukten jelas kanggo ngobati infeksi sing ana. Translate from Javanese to Chinese:

`text-davinci-003` output:

目前尚无明确的治疗方法可以治愈埃博拉病毒感染，但是研究人员已经开发出一种疫苗来预防埃博拉病毒引起的死亡。

`gpt-3.5-turbo` output:

目前为止，尚无明确的治疗方案可用于治疗埃博拉病毒感染，但如果接种一种疫苗可能会有所帮助，直至现在为止。

Answer:

虽然有一种实验性疫苗看似能够降低埃博拉病毒的死亡率，但迄今为止，还没明确证明任何药物适合治疗现有的感染。

Figure 24: A jv→zh example of the **FLORES\*** benchmark using **basic prompt** and the corresponding outputs under the **zero-shot** setting.

Input:

I want you to act as a machine translation expert for Javanese to Chinese .
Source sentence: Nalika ana siji vaksin pacoban katon bisa nyuda kematian Ebola, nganti saiki,
ora ana obat sing kabukten jelas kanggo ngobati infeksi sing ana.
You should repeat the source sentence in English.
You should provide the Chinese translation for the English source sentence .
You should step-by-step answer the request.
You should tell me the target translation in this format ' Target translation :'.

`text-davinci-003` output:

Source sentence: When there was a vaccine available to prevent death from Ebola, until now, there is no clear
evidence of a treatment to cure the infection.
Target translation: 当有一种疫苗可以预防埃博拉死亡时，直到现在，还没有清楚的证据表明有治疗感染的方法

`gpt-3.5-turbo` output:

Source sentence: Nalika ana siji vaksin pacoban katon bisa nyuda kematian Ebola, nganti saiki, ora ana obat sing
kabukten jelas kanggo ngobati infeksi sing ana.
English source sentence: When there is a vaccine trial that can help prevent Ebola deaths, until now, there is
no clear proven medicine to treat existing infections.
Chinese translation: 当有一种疫苗试验可以帮助预防埃博拉死亡时，到目前为止，没有明确的已证明的药物
来治疗现有的感染。
Target translation: 当有一种可以帮助预防埃博拉死亡的疫苗试验时，到目前为止，没有明确证明可以治疗现有感染的药
物。

Answer:

虽然有一种实验性疫苗看似能够降低埃博拉病毒的死亡率，但迄今为止，还没明确证明任何药物适合治疗现有的感染。

Figure 25: A jv→zh example of the **FLORES\*** benchmark using **XLT** and the corresponding outputs under the **zero-shot** setting.