# OpenReview forum: "Not All Languages Are Created Equal in LLMs: Improving Multilingual Capability by Cross-Lingual-Thought Prompting"
_EMNLP/2023/Conference — EMNLP 2023 Findings_

### Official Review · Reviewer_1a4U · 2023-07-28

**Soundness:** 3

**Excitement:**

3: Ambivalent: It has merits (e.g., it reports state-of-the-art results, the idea is nice), but there are key weaknesses (e.g., it describes incremental work), and it can significantly benefit from another round of revision. However, I won't object to accepting it if my co-reviewers champion it.

**Paper Topic And Main Contributions:**

The paper presents a study focusing on the multilingual capabilities of Large Language Models (LLMs). While these models have demonstrated impressive performance in multiple languages, the authors acknowledge significant variations in their effectiveness across different languages. To address this issue, they propose a method called Cross-Lingual-Thought Prompting (XLT) to systematically improve LLMs' multilingual capabilities. The main contribution related to new data resources and the central idea of XLT involves using a generic template prompt to stimulate cross-lingual and logical reasoning skills, ultimately enhancing the performance of LLMs on various tasks in diverse languages and particularly for low-resource languages.

**Questions For The Authors:**

Question A: In terms of ethical considerations, have you identified any potential biases or unfair advantages that could arise from the application of the XLT method in specific languages or tasks? It would be valuable to discuss any ethical implications associated with the proposed approach.

**Reasons To Accept:**

The paper introduces a new and innovative method, Cross-Lingual-Thought Prompting (XLT), to enhance the multilingual capabilities of Large Language Models (LLMs). This novel approach stimulates cross-lingual and logical reasoning skills, which has the potential to address the variability in LLM performance across different languages.

**Reasons To Reject:**

While the paper reports significant improvements in task performance with the XLT method, it could benefit from a more in-depth comparative analysis. For instance, comparing XLT against existing state-of-the-art methods for enhancing multilingual capability would provide a better understanding of its relative strengths and weaknesses.

**Reproducibility:**

4: Could mostly reproduce the results, but there may be some variation because of sample variance or minor variations in their interpretation of the protocol or method.

**Reviewer Confidence:**

3: Pretty sure, but there's a chance I missed something. Although I have a good feel for this area in general, I did not carefully check the paper's details, e.g., the math, experimental design, or novelty.

---

> ### Author Rebuttal · Authors · 2023-08-29
>
> Thanks for your positive comments!
>
> As you pointed out, our method can systematically improve LLMs' multilingual capabilities on various tasks in diverse languages, particularly for low-resource languages.
>
> Regarding your rejection reasons and questions:
> 1. As for the SOTA methods, the translate-English method [1, 2] is the most capable one to utilize LLMs. We have already included it as our baseline and included the original results from [1, 2] in Tables 7-13 for comparison. We will add this explanation in the final version.
> 2. About the ethical considerations, we directly utilize the API from OpenAI and follow the OpenAI cookbook. We think our template may not include additional biases since it mainly focuses on specific tasks rather than open-ended generation. We will add this discussion in the final version.
>
> [1] Language Models are Multilingual Chain-of-Thought Reasoners
>
> [2] MEGA: Multilingual Evaluation of Generative AI

---

### Official Review · Reviewer_eG1s · 2023-08-02

**Soundness:** 3

**Excitement:**

2: Mediocre: This paper makes marginal contributions (vs non-contemporaneous work), so I would rather not see it in the conference.

**Paper Topic And Main Contributions:**

This work proposes an optimized, generic, and language-independent prompt to enhance the multilingual capability of LLMs.

**Questions For The Authors:**

In essence, the work proposes a prompt template that enhances the behavior of the language model in multilingual tasks. However, besides a successful engineering effort, deeper insights are missing regarding which design factors are crucial for the model to succeed in multilingual problems. The ablation study and employed baselines are sufficient to demonstrate that the proposed prompt template improves the model's effectiveness. Nonetheless, it appears to be a trial-and-error-based approach. What I mean is that it would be intriguing if the authors provided some explanation, at least hypothetically, as to why the model performs better with this type of prompt.

Another aspect that is not covered is the exploitability of the proposal and its motivation. In reality, the very simplicity of the standard prompt on which the system works in the task is part of the problem that is intended to be solved. That is to say, in a real use case, I understand that we would have two types of users: the general user who expresses their needs without much detail or structure, or the expert user in a specific problem who gradually refines their way of communicating with the system. In the latter case, the user would never make use of the general prompt. What I mean is that it would be more interesting, rather than a fixed template, to have a more generic study on how prompts should be designed to cover rare languages. I understand that the optimal prompt template will always be task-dependent in the end. Is this correct?

In summary, I don't see any methodological objections in the article, although there is a lot of information about the datasets, metrics, etc., that cannot fit into the article. However, as a proposal itself, I believe that a prompt template is more suitable for a short paper, while a long paper should provide more generic results or insights into the problem.

**Reasons To Accept:**

Among the main conclusions, the authors state that prompting design factors such as instruction logic and word choice have explicit impacts on its effectiveness. This conclusion is very interesting and raises many questions.

The methodology and presentation is correct.

**Reasons To Reject:**

- Deeper insights are missing regarding which design factors are crucial for the model to succeed in multilingual problems.

- The exploitability of the proposal and its motivation is not clear.

- A prompt template is more suitable for a short paper, while a long paper should provide more generic results or insights into the problem.

**Reproducibility:**

4: Could mostly reproduce the results, but there may be some variation because of sample variance or minor variations in their interpretation of the protocol or method.

**Reviewer Confidence:**

4: Quite sure. I tried to check the important points carefully. It's unlikely, though conceivable, that I missed something that should affect my ratings.

---

> ### Author Rebuttal · Authors · 2023-08-29
>
> Thanks for your detailed comments!
>
> As you approved, our main conclusion and prompting design factors along with the methodology and presentation are correct and can enhance the multilingual capability of LLMs.
>
> Regarding your rejection reasons and questions:
> 1. About the deeper insights of our method, we try our best to explain them in the construction of XLT (Section 2.1). We follow the guidelines of the OpenAI cookbook, get the task information from existing literature, and leverage cross-lingual thinking and chain-of-thought thinking. The whole design is also similar to how humans solve multilingual problems. Moreover, CoT enhances the reasoning ability of LLMs, while our XLT improves the multilingual capabilities. We agree with your opinion that the insights of our XLT should be further explored, just as the dilemma CoT faces.
>
> 2. About the exploitability of our method, we agree with the statements of the two types of users. Therefore, the general user can easily leverage our template since XLT does not have any new design for general users when compared to basic prompts. They just need to fulfill six or seven simple metadata (e.g., task name and task language) and can achieve a significant performance improvement. As for the expert users, our template can be a good starting point for them to investigate task-specific or language-specific instructions, which is more delicate more basic prompts.

---

### Official Review · Reviewer_N1oa · 2023-08-03

**Soundness:** 3

**Excitement:**

3: Ambivalent: It has merits (e.g., it reports state-of-the-art results, the idea is nice), but there are key weaknesses (e.g., it describes incremental work), and it can significantly benefit from another round of revision. However, I won't object to accepting it if my co-reviewers champion it.

**Paper Topic And Main Contributions:**

This paper introduces a generic language-independent prompt called “cross-lingual thought prompting” (XLT) for enhancing the multilingual capabilities of large language models (LLMs). XLT instructs model to repeat the task from a random language to English, think in English, and finally respond in its original language. Experimental results show improvements in multilingual tasks, reducing performance differences between languages.

**Questions For The Authors:**

- The explanation provided in Section 2.2 is vague. Can you provide more details or examples to clarify "demonstrations"?
- Could you provide a clearer understanding of "Language Democratization"? How does prompting the model to think in English facilitate democratization?


**Reasons To Accept:**

- This paper introduces XLT, a straightforward yet powerful method that enhances multilingual capabilities in large language models (LLMs). It achieves impressive improvements on several multilingual NLP tasks.
- The article explores various variants of prompts, confirming the effectiveness of XLT.

**Reasons To Reject:**

- All experiments are based on two models: "text-davinci-003" and "gpt-3.5-turbo", where these two models are supposed to be very similar (compared to other LLMs such as LLaMA, Vicuna, GLM, BLOOM etc). The author even analysis wording such as whether to use "translate" or "retell". Which might be very model specific. Despite this limitation, the author asserts that the prompt template is meant to enhance the multilingual abilities of "LLMs", an assertion yet untested on other LLMs. A study on multilingual LLMs is expected to include experiments on models like BLOOM (BLOOMZ) or mT5 (mT0).
- Key information missing: the paper does not specify how the model output is extracted as "label", or how it handles missing or improperly formatted responses. The baseline prompt does not contains formatting instruction but XLT contains. Besides, there is no ablation results on zero-shot format (i.e. remove the format instruction). Hence, we do not know how much improvements are bring by this formatting instruction.
- Unnatural baselines. There are three baselines, The first two baselines are in English despite the tasks being multilingual. The third baseline translates everything into English, introducing translationese. A more natural baseline would involve the prompt being in the same language as the task.
- The term "Language Democratization" is introduced in the paper with a scoring system to gauge the degree of democratization. However, the definition and implication of this term are unclear. It is hard to agree with the notion that transferring questions to English qualifies as democratization, as this gives English an inflated importance. On the other hand, even for a task, the samples in different language might from different sources therefore have different difficulties, I cannot get the point why this score can reflect "Language Democratization".

**Reproducibility:**

4: Could mostly reproduce the results, but there may be some variation because of sample variance or minor variations in their interpretation of the protocol or method.

**Reviewer Confidence:**

4: Quite sure. I tried to check the important points carefully. It's unlikely, though conceivable, that I missed something that should affect my ratings.

**Typos Grammar Style And Presentation Improvements:**

- Table 1 displays an error where the XLT value of 23.9 is highlighted, even though CoT's 25.7 is higher.
- Extra spaces can be found after the subheadings on lines 149, 157, 164, 175, 182, 190.
- Sentences 391 to 404 use the ';' symbol to connect points, while sentences 446 to 456 use '.' for the same purpose. Consistency in the choice of symbol is recommended.
- The captions for Tables 5 and 6 are positioned above the tables, which should ideally be placed below.

---

> ### Author Rebuttal · Authors · 2023-08-29
>
> Thanks for your meaningful comments!
>
> As you stated, our straightforward yet powerful XLT prompting method can enhance the multilingual capabilities of large language models.
>
> Regarding your rejection reasons and questions:
> 1. Your concerns about the applicability of our method are meaningful. Our proposed cross-lingual thought is not dependent on any specific large language models. We choose the two popular models: text-davinci-003 and gpt-turbo-3.5 due to resource limitations. We will follow your suggestions and add experiments on BLOOMZ or mT0 to verify the applicability of our method in our final version.
>
> 2. About the details of label extraction, the illustrations can be found in the Appendix. Similar to many related works, we in practice found these LLMs have superior instruction-following ability, so they can follow our instructions to respond to the query in the given format. Therefore, we just intercept the part after “Answer format:” for score calculation. In the meantime, the formatting instruction is essential for our method to extract the answer from the intermediate cross-lingual thoughts. If we remove that, it is not a trivial work to get the final answer. We will add this explanation in the final version.
>
> 3. About the baselines, researchers in [1, 2] have investigated English prompting and native-language prompting, and they reached the same conclusion that English is better. According to those results, we omit this baseline. We will add this explanation in the final version.
>
> 4. Our experiments are conducted on the tasks (MGSM, XCOPA, XNLI, PAWS-X, MKQA) whose instances are semantic equivalents from different languages. We use the democratization score to indicate the language capability gap under the same semantic meaning. In detail, we calculate the score attained by different languages relative to the best performance among all languages (line 345) rather than just English. We think that if task performance across languages is closer, the gap between languages becomes smaller, thus approaching the notion of democracy. We will add this explanation in the final version.
>
> 5. About the examples of demonstrations, we utilize the same template to generate corresponding output in a zero-shot manner. One demonstration consists of the template and the zero-shot output. More examples of demonstrations can be found in the Appendix.
>
> [1] Language Models are Multilingual Chain-of-Thought Reasoners
>
> [2] MEGA: Multilingual Evaluation of Generative AI

---

### Meta-Review · Area_Chair_ctPY · 2023-09-19

**Recommendation:** 3

**Metareview:**

This paper investigates prompting an LLM to perform "cross-lingual" chain-of-thought reasoning, in which it first translates the task to English, reasons in English, and then answers in the original language. Experiments show that this yields improvements on several tasks.

The reviewers appreciated that the approach is intuitive, and effective for the models used in experiments.

However, they also had several concerns, including:

* They would have liked to see experiments on more than just two models, and particularly would have liked to see experiments on models whose training was focused on multilinguality (rather than just models main focus is English).
* They felt the baselines were unnatural/unfair, and the paper omitted the most natural baseline of just prompting the model in the target language.
* They felt that the paper lacks the level of generalizability or analysis necessary for a long paper.

---

### Decision · Program_Chairs · 2023-10-07

**Decision:**

Accept-Findings

**Comment:**

This paper investigates prompting an LLM to perform "cross-lingual" chain-of-thought reasoning, in which it first translates the task to English, reasons in English, and then answers in the original language. Experiments show that this yields improvements on several tasks.

The reviewers appreciated that the approach is intuitive, and effective for the models used in experiments.

However, they also had several concerns, including:

* They would have liked to see experiments on more than just two models, and particularly would have liked to see experiments on models whose training was focused on multilinguality (rather than just models main focus is English).
* They felt the baselines were unnatural/unfair, and the paper omitted the most natural baseline of just prompting the model in the target language.
* They felt that the paper lacks the level of generalizability or analysis necessary for a long paper.